# Value of SUV_max_ for the Prediction of Bone Invasion in Oral Squamous Cell Carcinoma

**DOI:** 10.3390/biology9020023

**Published:** 2020-02-02

**Authors:** Stephanie A. Stalder, Paul Schumann, Martin Lanzer, Martin W. Hüllner, Niels J. Rupp, Martina A. Broglie, Grégoire B. Morand

**Affiliations:** 1Department of Otorhinolaryngology - Head and Neck Surgery, University Hospital Zurich, 8091 Zurich, Switzerland; stephistalder@gmx.ch (S.A.S.); martina.brogliedaeppen@usz.ch (M.A.B.); 2Faculty of Medicine, University of Zurich, 8006 Zurich, Switzerland; 3Department of Cranio-Maxillo-Facial and Oral Surgery, University Hospital Zurich, 8091 Zurich, Switzerland; paul.schumann@usz.ch (P.S.); martin.lanzer@usz.ch (M.L.); 4Department of Nuclear Medicine, University Hospital Zurich, 8091 Zurich, Switzerland; martin.huellner@usz.ch; 5Department of Pathology and Molecular Pathology, University Hospital Zurich, 8091 Zurich, Switzerland; niels.rupp@usz.ch

**Keywords:** carcinoma, squamous cell, positron emission tomography, fluorodeoxyglucose F18, bone, tumor hypoxia

## Abstract

In advanced oral squamous cell carcinoma (OSCC), accurate planning of surgical resection and reconstruction are crucial for outcome and postoperative function. For OSCC close to the maxilla or mandible, prediction of bone invasion is necessary. The aim of this study was to examine whether metabolic tumor imaging obtained by fluorodeoxyglucose positron emission tomography (FDG-PET) could enhance preoperative predictability of bone invasion. We performed an analysis of 84 treatment-naïve OSCCs arising from gum (upper and lower), hard palate, floor of mouth, and retromolar trigone treated at the University Hospital Zurich, Switzerland, who underwent wide local excision with free flap reconstruction between 04/2010 and 09/2018 and with available preoperative FDG-PET. Prediction of bone invasion by metabolic tumor imaging such as maximum standardized uptake value (SUV_max_) was examined. On definitive histopathology, bone invasion was present in 47 of 84 cases (56%). The probability of bone infiltration increased with a higher pretherapeutic SUV_max_ in an almost linear manner. A pretherapeutic SUV_max_ of primary tumor below 9.5 ruled out bone invasion preoperatively with a high specificity (97.6%). The risk of bone invasion was 53.6% and 71.4% for patients with SUV_max_ between 9.5–14.5 and above 14.5, respectively. Patients with bone invasion had worse distant metastasis-free survival compared to patients without bone invasion (log-rank test, *p* = 0.032). In conclusion, metabolic tumor imaging using FDG-PET could be used to rule out bone invasion in oral cancer patients and may serve in treatment planning.

## 1. Introduction

Oral squamous cell carcinoma (OSCC) is an aggressive malignancy characterized by local invasiveness and high propensity to lymph node metastases [1]. OSCC is the sixth most common cancer worldwide and the most common site of malignancy in the head and neck [2]. Risk factors for OSCC include exposure to extrinsic carcinogens such as tobacco, alcohol, and betel nut [1,2]. The incidence of OSCC varies among geographical regions accordingly, with a high incidence in Melanesia, South Central Asia, Australia, and Europe [3]. The unchanging survival in patients with OSCC underscores the need for better prognostic tools, as recently reported in the 8th edition of American Joint Committee on Cancer staging system [4].

OSCC is primarily amenable to surgery and requires wide local excision with macroscopical margins of usually at least 1.0 cm. In advanced cases, wide local excision leads to a substantial defect, which has to be reconstructed with a local or free flap [5].

If the tumor is close to the bone, but without bone invasion, it may be resected together with the periosteum. In select cases, a marginal mandibulectomy may be required as well. Reconstruction with a fascio-cutaneous or musculocutaneous soft tissue flap is then sufficient [2].

If the tumor has invaded the bone, a segmental mandibulectomy is often required. This requires a reconstruction using an osteo(myo)cutaneous free flap. Accurate preoperative assessment of bone invasion is hence an important prerequisite for reconstruction planning and patient consent [2].

While clinicians have traditionally relied on physical examination, including triple endoscopy, some have examined the accuracy of cross-sectional imaging such as computed tomography (CT) and/or magnetic resonance imaging (MRI) for bone invasion [6,7,8,9,10]. It turned out, however, that these modalities have limited diagnostic accuracy [10].

Recent studies suggest that metabolic tumor imaging with 18-fluorodeoxyglucose positron emission tomography (FDG-PET) could be useful for estimating the aggressiveness of a particular tumor [11,12,13]. Here, the most commonly used metabolic parameter is the maximum standardized uptake value (SUV_max_).

Whether metabolic tumor imaging could add additional information for the prediction of bone invasion has not been investigated yet. Therefore, the goal of our study was to examine if FDG uptake could increase the predictability of bone invasion in OSCC.

## 2. Results

### 2.1. Patient and Tumor Characteristics

A total of 84 treatment-naïve patients with curative intent suffering from advanced OSCC from the upper and lower gum, retromolar trigone, floor of mouth, and hard palate with available pretherapeutic PET/CT or PET/MRI were included in this study (Table 1). The mean age at diagnosis was 67.3 (SD 11.0) years. There was a clear male predominance with 52 (61.9%) male and 32 (38.1%) female patients. Seventy-four (88.1%) had squamous cell carcinomas close to/with invasion of the mandible, and 10 patients (11.9%) close to/with invasion of the maxilla. Most patients (63.1%) had pT4 tumors while 33.3% and 3.6% of patients had pT1-pT2 and pT3 tumors, respectively. Nodal status was positive in 43 patients (51.2%), of which 17 (20.2%) were staged with pN1, 22 (26.2%) with pN2a-pN2b, and 4 (4.8%) with pN2c-pN3 categories.

The median pretherapeutic SUV_max_ was 14.10 (IQR 10.63–17.5) for the whole cohort. The median follow-up time for all patients was 20 months (IQR 9.25–36.25). A total of 46 patients (54.8%) were treated with adjuvant radio(chemo)therapy (Table 1).

### 2.2. Patients with Bone Invasion Had Higher SUV_max_ of Primary Tumor

As depicted in Table 1, there were 47 patients with bone invasion and 37 without bone invasion. Comparisons between the two groups showed a trend towards older age in patients with bone invasion. As expected, there were significantly more patients classified as pT4 in the bone invasion group (chi-squared test, *p* < 0.001). The SUV_max_ of the primary tumor was significantly higher in the bone invasion group (Mann–Whitney U test, *p* = 0.015) (Figure 1).

### 2.3. Prediction of Mandibular Infiltration Using Clinical Examination, Computed Tomography (CT), and Magnetic Resonance Imaging (MRI)

The different modalities were compared against each other with regard to sensitivity and specificity for predicting bone invasion. Bone invasion assessed by definitive histopathological analysis was used as the gold standard. As shown in Table 2, clinical examination had limited sensitivity and specificity. CT had a poor sensitivity but a rather good specificity, meaning that a positive CT finding was very likely to predict bone invasion. MRI had a poorer diagnostic accuracy than CT for bone invasion.

### 2.4. Prediction of Bone Invasion Using Metabolic Tumor Imaging

Various cutoff values for FDG uptake parameters were tested. Using receiver operating characteristic (ROC) curves, the best potential cutoff value for the pretherapeutic SUV_max_ of primary tumor was determined to be 9.5 (Figure 2; area under the curve (AUC) 66.5% (95% CI 53.9%–79.1%), *p* = 0.015). The risk for bone invasion was also evaluated in ordinal fashion. For patients with pretherapeutic SUV_max_ < 9.5, the risk for mandibular invasion was very low with 9.1%. It was 53.6% for patients with SUV_max_ between 9.5 and 14.5, while patients with a strong metabolically active primary (SUV_max_ > 14.5) had a very high risk (71.4%) for bone invasion. The risk for bone invasion increased with a higher pretherapeutic SUV_max_ in an almost linear manner, as depicted in Figure 3.

### 2.5. Enhanced Diagnostic Accuracy Using Metabolic Imaging

We integrated the metabolic imaging parameter SUV_max_ of primary tumor (with a cutoff value of 9.5) and calculated the sensitivity and the specificity for bone invasion. The sensitivity and specificity for metabolic tumor imaging were 97.6% and 31.2%, respectively.

In summary, computed tomography had a good specificity, while metabolic tumor imaging demonstrated a high sensitivity with a remarkably low number of false negatives (Table 2).

### 2.6. Survival Outcomes

We performed Kaplan–Meier analysis to examine relative survival according to bone invasion. Local recurrence-free survival (Figure 4, Panel A, log-rank, *p* = 0.855) and regional recurrence-free survival (Figure 4, Panel B, log-rank, *p* = 0.536) were similar for both groups. Patients with bone invasion showed a worse distant metastasis-free survival (Figure 4, Panel C, log-rank, *p* = 0.032). Disease-specific survival was similar among the two groups (Figure 4, Panel D, log-rank, *p* = 0.144).

## 3. Discussion

This retrospective study evaluates the predictability of bone invasion by FDG-PET parameters of OSCC in a cohort of 84 patients. Primary tumor SUV_max_ > 14.5 showed a high probability for bone invasion, while the risk for bone invasion was low for primaries with SUV_max_ < 9.5. This study showed that considering the SUV_max_ of the primary tumor could increase the diagnostic accuracy for prediction of bone invasion by OSCC. Using such metabolic tumor imaging, the sensitivity of preoperative imaging could be considerably increased, mainly by reducing the number of false negatives. Practically, our findings imply that bone invasion can be largely ruled out if SUV_max_ of the primary OSCC tumor is <9.5 with a sensitivity of 97.6%. Such a finding has a direct clinical applicability for treatment planning, considering complexity and complication rate for bone flaps compared to soft tissue flaps. Bone flaps require significant planning time for presurgical 3D planning for optimal bone and dental rehabilitation. They also have a higher failure rate than fascio- or musculocutaneous flaps [14].

Previous studies have examined the diagnostic accuracy of several imaging modalities for the prediction of bone invasion [6,7,9,10]. A metaanalysis by Qiao et al. reported the diagnostic accuracy of eight different diagnostic modalities. They showed a similar pooled accuracy of the examined modalities with heterogenous and inconsistent data from each examined study. Regarding CT and MRI, their study found similar values for sensitivity and specificity as reported in our study [10], with a slightly better diagnostic accuracy for MRI than in our study. Qiao et al. did not report the diagnostic accuracy of the clinical examination and/or triple endoscopy. The latter is, however, quite difficult in clinical practice and not reliable enough. In our study, it was inferior to both CT and MRI. 

When comparing CT and MRI, the latter seems to be more vulnerable for misinterpretation, owing to metal-related artifacts, while the former is more dependent on tumor origin and dentate status. CT, however, showed a quite good specificity. Therefore, in case of positive bone invasion on CT, the latter is very likely to confirm on histopathological analysis. MRI had a rather poor diagnostic accuracy for bone invasion in our study. 

For metabolic tumor imaging, an SUV_max_ of primary tumor <9.5 had a very high sensitivity, that is, allowing one to rule out the presence of bone invasion. This can be used nicely in complement to the CT findings. 

FDG-PET seems to be slightly better than cross-sectional imaging alone, while a recent study by Sekine et al. showed that there is no overall difference between PET/CT and PET/MR in the detection of bone invasion [15]. False-positive findings of bone marrow invasion are reported for FDG-PET. They were significantly higher in edentulous patients than in dentate patients [7]. One possible explanation is the presence of periodontitis, which leads to dental loss and higher FDG uptake due to inflammatory process [7]. This is generally a drawback of FDG-PET, which has a very good sensitivity but is prone to false-positive results due to, e.g., inflammation [16].

The role of the teeth in conditioning the bone invasion of the mandible and maxilla remains controversial. Edentulous patients are thought by some to be more prone to bone infiltration than dentate patients, as teeth represent a relative barrier against tumor infiltration, and tooth loss leads to reduction of the height and occlusal surface of the mandible [15]. Some authors suggest, however, that the tumor can gain access to the dentate mandible through the periodontal membrane and invade more easily. Finally, others found that bone invasion occurs mainly at the point of junction of the attached gingival and reflected mucosa of the alveolar bone in both dentate and edentulous patients [2,16]. 

As shown in previous reports, SUV_max_ seems to be the metabolic parameter showing the highest reliability among all metabolic parameters [12,13]. It represents the single voxel with the most intense FDG uptake of a lesion and can be assessed easily and with high reproducibility. Hence, the SUV_max_ is a parameter that is used in daily clinical practice in every PET center worldwide. 

In previous studies, we described that metabolic tumor imaging can be used as a surrogate to determine tumor aggressiveness [12,13]. Aggressive tumors undergo many genotypic and phenotypic changes through epithelial-mesenchymal transition (EMT) and cancer stem cells (CSC) enrichment to acquire invasive and metastatic properties [17]. Among others, these changes results in the expression of CD44, which allows cells to invade more deeply the surrounding tissue and enhances the glycolytic phenotype of cancer cells that are exposed to hypoxia [18]. The metabolic changes in hypoxic tumors induce tumor cells to increasingly metabolize glucose through glycolysis rather than the oxygen-dependent Krebs cycle (Warburg effect) [19]. An overall enhanced glucose consumption is observed in cancer cells with an increase of glucose transporter 1 (GLUT1) expression [20,21]. The glucose uptake of the tumor increases, which can be quantified using the SUV_max_ of FDG-PET. As a result, the SUV_max_ may be used as a surrogate marker for tumor aggressiveness, because it is positively correlated with tumor hypoxia.

Our study used different PET scanners with partly different reconstruction methods. However, the SUV_max_ is a standardized and comparably stable parameter. Our study did not stratify tumors by subsites in the oral cavity, although we did exclude OSCC tumors without any close anatomical relationship with the bone (oral tongue, buccal mucosa). It is known that tumors arising in the upper and lower gum or floor of the mouth are more prone to bone invasion than, e.g., tongue tumors [15]. It would have been exceedingly difficult, if not impossible, to create a truly comparable cohort of patients with or without bone invasion. Instead, we chose to include all OSCCs close to the bone with free flap reconstruction, thus avoiding selection bias. Further, we did not analyze the association between bone turnover markers and metabolic tumor imaging parameters. Some studies indeed reported the possibility of measuring bone turnover markers (such as bone sialoprotein) to increase the predictability of bone metastases [22,23]. It would have been very interesting to see if similar observations could be made for OSCC, bone invasion, and metabolic tumor imaging.

In conclusion, we showed that metabolic tumor imaging could be a useful clinical tool for ruling out bone infiltration in OSCC, simply by considering the SUV_max_. This can easily be accomplished in clinical routine.

## 4. Materials and Methods

### 4.1. Study Population

After Ethics Review Board approval by the *Kantonale Ethikkomission Zürich* (protocol number 2016-01799, including amendment of 14 December 2018), all patients treated for squamous cell carcinoma of the oral cavity (OSCC) between 1 April 2010 and 1 September 2018, at the Department of Otorhinolaryngology – Head and Neck Surgery and the Department of Maxillofacial Surgery of the Zurich University Hospital, Switzerland, were retrospectively assessed. Study methods were carried out in accordance with the relevant guidelines and regulations. Informed consent of all enrolled patients was obtained. All patients with advanced oral cancer undergoing wide local excision and neck dissection and reconstruction were included. Only OSCC arising from the floor of mouth, upper and lower gum, hard palate, and retromolar trigone were included. Primary tumors from the lip, oral tongue, and the buccal mucosa were excluded. Primary study outcomes were predictors of bone invasion to the mandible or the maxilla. To determine predictors of bone invasion, clinical, radiological, and metabolic tumor imaging factors were assessed. Histopathological proof of bone invasion served as the gold standard. Secondary study outcomes were local-recurrence-free, regional-recurrence-free, distant-metastasis-free, and disease-specific survival.

Inclusion criteria were available pretherapeutic FDG-PET/CT or FDG-PET/MR images and treatment-naïve patients with curative intent. Patients treated surgically after induction chemotherapy were excluded.

All patients were staged according to the *Union Internationale Contre le Cancer* (UICC), TNM staging for head and neck cancer, 7th edition, 2010. We chose to use the 7th edition and not restage all patients with the UICC 8th edition, as the clinical decisions (imaging, surgery, adjuvant treatment) were based on the 7th edition of the TNM staging, which was in use at the time of treatment in all subjects [24]. After full medical history, physical examination, triple endoscopy with biopsy, and imaging with FDG-PET were performed. All patients were presented and discussed at the local multidisciplinary tumor board. Frozen sections were used intraoperatively to assure free mucosal and soft tissue margins of the surgical resection specimen. All patients had negative margins upon final pathology (R0).

Detailed data on age, gender, smoking, drinking habits, clinical and pathological tumor stage, bone invasion, local and regional recurrence, distant metastasis, disease-specific survival, and overall survival were obtained. Smoking was defined as a current daily consumption of cigars or cigarettes. Alcohol consumption was defined as a daily intake of more than 20 g of ethanol for at least five days a week. The study cohort was then divided into two groups according to the presence or absence of bone invasion defined by histopathology.

### 4.2. FDG-PET/CT or -/MR Image Acquisition

Patients were injected with a standardized dose of 3.5 MBq of 18-fluorodeoxyglucose (FDG) per kilogram body weight after fasting for at least four hours. All patients had a blood glucose level below 12 mmol/L before imaging. Patients were instructed to remain in a lying or recumbent position and silent for 50–60 min to minimize muscular FDG uptake in the period between FDG injection and image acquisition. Patients were also kept warm prior to tracer injection and throughout the uptake period to diminish FDG accumulation in brown adipose tissue. All patients received either iodinated or gadolinium-based contrast medium. An integrated Discovery VCT PET/CT system (GE Healthcare, Waukesha, WI, USA), a Discovery PET/CT 690 (GE Healthcare), or a hybrid PET/MRI system (Signa PET/MR, GE Healthcare) was used for image acquisition.

### 4.3. Image Assessment

Selected parameters of tumor FDG metabolism were recorded under supervision of a dually board-certified nuclear physician and radiologist and included SUV_max_ of the primary tumor. SUV_max_ was calculated automatically using a standard formula (maximum activity in region of interest ÷ (injected dose × body weight)). A correct referencing of FDG uptake was ensured through side-by-side reading of the corresponding CT or MR images of the tumor in the axial, coronal, and sagittal plane. Borders of regions of interest (ROI) were set by manual adjustment to exclude adjacent physiologic FDG-avid structures. A written report by a dually board-certified nuclear medicine physician/radiologist was available for all FDG-PET/CT or -/MR images.

### 4.4. Statistical Analysis

For continuous variables, distribution was evaluated for normality according to Gauss’ theorem. For normally distributed variables (age), mean and standard deviations are given, and comparison among study groups was done using the t-test. For non-normally distributed variables (smoking, SUV_max_ primary tumor, follow-up time), median and interquartile range (IQR) are given. To compare distribution among samples, the nonparametric Mann–Whitney U test was used for two samples. Binary variables were associated in contingency tables using the two-tailed Pearson chi-squared test. Bone invasion was defined by histopathology. The sensitivity and specificity of clinical examination, morphological cross-sectional imaging (MR and CT) and combined modality with FDG-PET metabolic parameter were calculated according to Bayes’ theorem. Receiver operating characteristic (ROC) curves were used to select the best cutoff value for SUV_max_ to predict high risk for bone invasion with the 95% confidence interval provided (95% CI). Survival curves were built according to Kaplan–Meier, and the log-rank test was used to compare factors. A *p*-value lower than 0.05 was considered to indicate statistical significance. Statistical analyses were performed using SPSS^®^ 25.0.0.1 software (IBM^®^, Armonk, NY, USA).

## 5. Conclusions

In this retrospective study, we showed that metabolic tumor imaging could be a useful clinical tool for ruling out bone invasion in OSCC, simply by considering the SUV_max_. This can easily be accomplished in clinical routine due to good accessibility in many facilities. Treatment planning could be more concise because of good predictability of whether bone invasion is present or not considering SUV_max_ of the primary. Future studies shall examine the performance of SUV_max_ in a prospective setting as a help for treatment and reconstruction planning.

## Figures and Tables

**Figure 1 biology-09-00023-f001:**
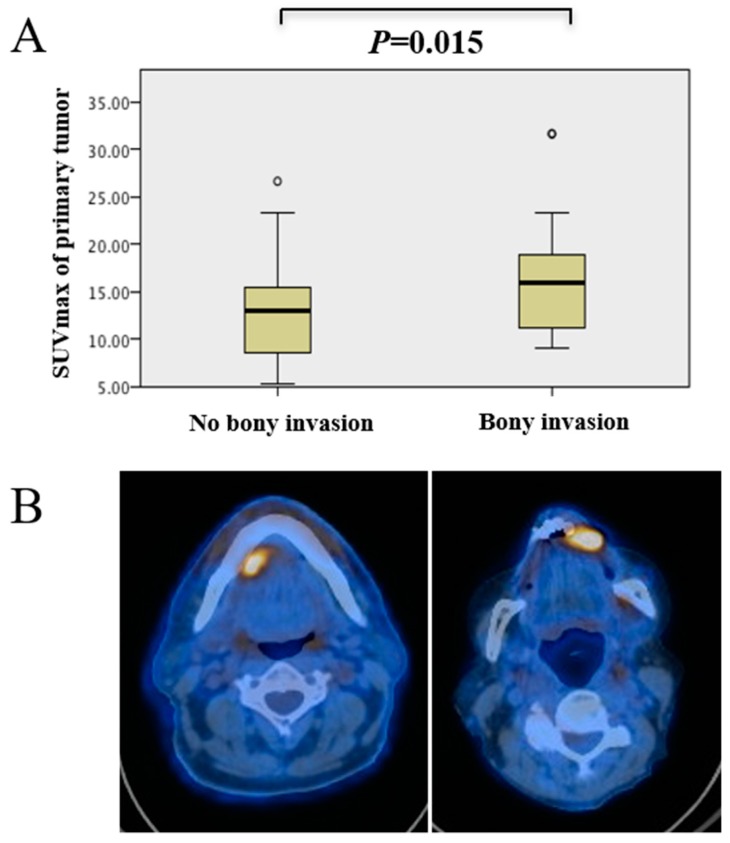
(**A**) Correlation between maximum standardized uptake value (SUV_max_) of primary tumor and bone invasion. Oral squamous cell carcinoma (OSCC) with bone invasion showed a significantly higher SUV_max_ (Mann–Whitney U Test, *p =* 0.015). (**B**) Representative axial fused FDG-PET/CT images showing, on the left, a tumor with a low SUV_max_ (7.3). Histopathological analysis showed no bone invasion. On the right, the SUV_max_ was 21.8, and the tumor showed bone invasion.

**Figure 2 biology-09-00023-f002:**
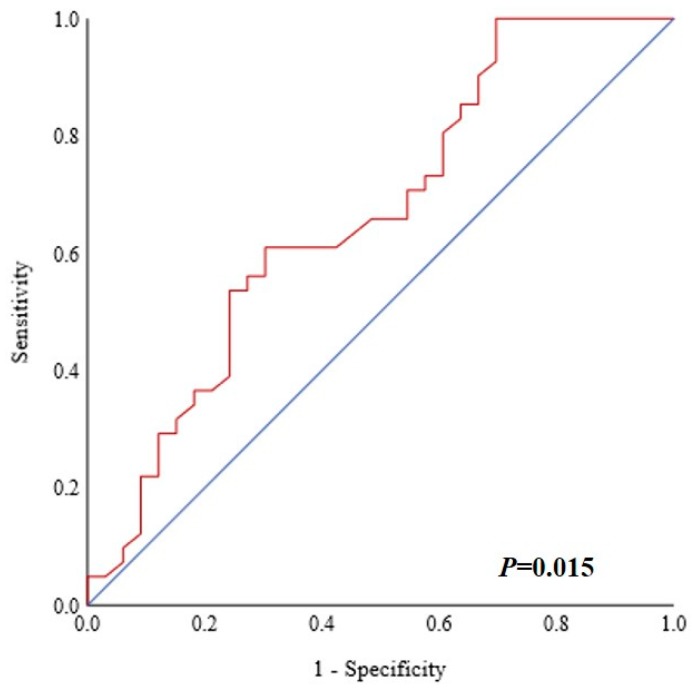
Receiver operating characteristic (ROC) curve showing the maximum standardized uptake value (SUV_max_) and bone invasion. The best cutoff was determined to be 9.5. The area under the curve (AUC) was 66.5% (95% CI 53.9%–79.1%) (*p* = 0.015).

**Figure 3 biology-09-00023-f003:**
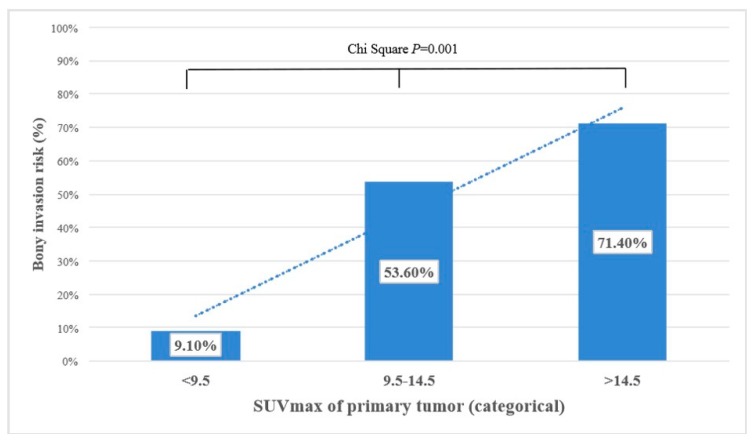
Categorical representation of maximum standardized uptake value (SUV_max_) and bone invasion risk. Oral squamous cell carcinoma (OSCC) with low SUV_max_ had a very low risk for bone invasion, while the risk increased in an almost linear manner with increasing SUV_max_ (Pearson chi-squared test, *p* = 0.001).

**Figure 4 biology-09-00023-f004:**
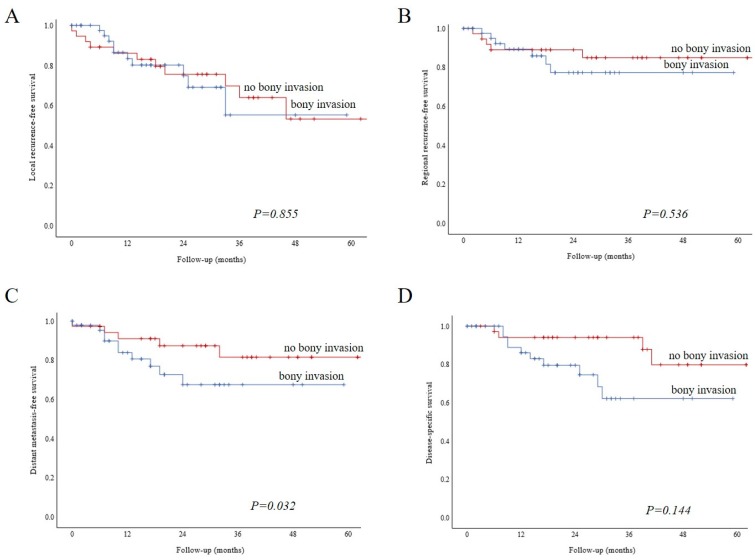
Kaplan–Meier analysis showing relative survival according to bone invasion. Local recurrence-free survival (**A**, log-rank, *p =* 0.855) and regional recurrence-free survival (**B**, log-rank *p =* 0.536) were similar for both groups. Patients with bone invasion had a worse distant metastasis-free survival. (**C**, Log-rank, *p =* 0.032). Disease-specific survival was similar among the two groups (**D**, log-rank, *p* = 0.144).

**Table 1 biology-09-00023-t001:** Patient Demographics and Clinical Characteristics.

Variable		All Patients No. of Patients = 84	Bone Invasion No. of Patients = 47	No Bone Invasion No. of Patients = 37	*p* Value ^1^ B vs. nB
***Age***					
**Years**	Mean (SD)	67.3 (11.0)	69.3 (11.5)	64.7 (9.9)	0.055
***Gender***					
**Male**	No. (%)	52 (61.9%)	26 (55.3%)	26 (70.3%)	
**Female**	No. (%)	32 (38.1%)	21 (44.7%)	11 (29.7%)	0.182
***Smoking***	Yes (%)	43 (51.2%)	21 (44.7%)	22 (59.5%)	
	No (%)	41 (48.8%)	26 (55.3%)	15 (40.5%)	0.195
***Alcohol consumption***	Yes (%)	42 (50.0%)	19 (40.4%)	23 (62.2%)	
	No (%)	42 (50.0%)	28 (59.6%)	14 (37.8%)	0.128
***cT-classification***					
**T1-T2**	No. (%)	22 (26.2%)	8 (17%)	14 (37.8%)	
**T3**	No. (%)	5 (6.0%)	0	5 (13.5%)	
**T4**	No. (%)	57 (67.8%)	39 (83%)	18 (48.7%)	0.007
***pT-classification***					
**T1-T2**	No. (%)	28 (33.3%)	0	27 (73%)	
**T3**	No. (%)	3 (3.6%)	0	3 (8.1%)	
**T4**	No. (%)	53 (63.1%)	47 (100%)	7 (18.9%)	<0.001 *
***pN-classification***					
**N0**	No. (%)	41 (48.8%)	18 (38.3%)	23 (62.2%)	
**N1**	No. (%)	17 (20.2%)	8 (17%)	9 (24.3%)	
**N2a-b**	No. (%)	22 (26.2%)	18 (38.3%)	4 (10.8%)	
**N2c-N3**	No. (%)	4 (4.8%)	3 (6.4%)	1 (2.7%)	0.023
***FDG-PET***					
**SUV_max_ primary tumor**	Median (IQR)	14.1 (10.6–17.5)	15.9 (11.1–18.9)	13.0 (8.1–16.0)	0.015 *

^1^*T*-Test for normally distributed variables. Mann–Whitney U Test for non-normally distributed variables, 2-sided Pearson chi-squared test for categorical variables. SUV_max_: maximum standardized uptake value. *p*-value for null hypothesis; * statistically significant.

**Table 2 biology-09-00023-t002:** Comparative table of prediction of bone invasion compared to histopathological standard.

Variable	Histopathological Examination		
***Clinical examination***	TP	FP	Sensitivity
	40	19	67.8%
	FN	TN	Specificity
	7	18	72.0%
***Computed tomography (CT)***	TP	FP	Sensitivity
	13	1	72.2%
	FN	TN	Specificity
	5	11	91.6%
***Magnetic resonance imaging (MRI)***	TP	FP	Sensitivity
	28	6	73.3%
	FN	TN	Specificity
	8	19	76.0%
***SUV_max_ primary tumor < 9.5***	TP	FP	Sensitivity
	41	22	97.6%
	FN	TN	Specificity
	1	10	31.2%

TP: true positive. FP: false positive. FN: false negative. TN: true negative.

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
