# Peer review of "Value of SUVmax for the Prediction of Bone Invasion in Oral Squamous Cell Carcinoma"

_biology, 2020, doi:10.3390/biology9020023_

Round 1
Reviewer 1 Report
This is an interesting retrospective study that aimed to examine whether metabolic tumor imaging could enhance preoperative predictability of bone invasion. In general the manuscript is well written; please check throughout the text for grammar and spelling errors. Please in the Abstract (line 24) specify that metabolic tumor imaging was obtained through FDG-PET and add the number of the patients (N=84) included in the study. In the introduction (line 47 to 56), please place appropriate references at the end of the statements and not all the references at the end of the paragraph “[3–8]”. Methods section and statistical analysis section are clearly described. The figures and tables are clear and useful. I would suggest to include more discussion on the possible association between bone turnover markers and imaging techniques to increase the predictability of bone metastases (e.g. Uccello M, et al J Cancer Res Ther. 2011; Ferreira A, et al Bonekey Rep. 2015). Finally, I would include a statement in the conclusion paragraph on the future direction and possible application of the results.
Author Response
Reviewer 1:
This is an interesting retrospective study that aimed to examine whether metabolic tumor imaging could enhance preoperative predictability of bone invasion. In general the manuscript is well written; please check throughout the text for grammar and spelling errors. Please in the Abstract (line 24) specify that metabolic tumor imaging was obtained through FDG-PET and add the number of the patients (N=84) included in the study. In the introduction (line 47 to 56), please place appropriate references at the end of the statements and not all the references at the end of the paragraph “[3–8]”. Methods section and statistical analysis section are clearly described. The figures and tables are clear and useful. I would suggest to include more discussion on the possible association between bone turnover markers and imaging techniques to increase the predictability of bone metastases (e.g. Uccello M, et al J Cancer Res Ther. 2011; Ferreira A, et al Bonekey Rep. 2015). Finally, I would include a statement in the conclusion paragraph on the future direction and possible application of the results.
Thank you for your encouraging comments.
Abstract: line 24: FDG-PET specified and number of patients specified.
Introduction (line 47-56): References were updated and placed at the end of each corresponding statement. References numbers changed completely after integration of the new references suggested by both reviewers
Discussion: the discussion was extended to integrate the possibility of bone turnover markers. Both suggested studies are now cited. This is indeed a very interesting point that shall be examined in a cohort of patients with available blood samples.
Conclusion: a statement has been added to the conclusion regarding the future direction and application of the results of this study.
Spell check repeated.

Reviewer 2 Report
Manuscript title: “Value of SUVmax for the Prediction of Bone Invasion 2 in Oral squamous cell Carcinoma”
In this nice work, the Authors investigated the use of metabolic tumor imaging with 18-fluorodeoxyglucose positron emission tomography (FDG-PET) to predict the bone invasion in OSCC. In particular, clinical, radiological and metabolic tumor imaging factors were assessed in 84 patients, using pathological data as gold standard. The maximum standardized uptake value (SUVmax), calculated using the formula: max. activity in region / (injected dose × body weight).
The results were interesting, showing that a value of SUVmax below 9.5 ruled out bone invasion preoperatively with a high specificity (97.6%).
The techniques used were appropriate and described with plenty details. Overall, this is a well-designed study with rigorous methods. The discussion is well-balanced, and the statements are supported by the data. The study is on a timely subject in view of increasing interest about the identification of new prognostic and predictive markers in oral oncology.
Only minor language corrections should be necessary.
Furthermore, I suggest to expand the Introduction section by adding some considerations related to the epidemiology and prognosis of OSCC. In particular, the unchanging survival in patients with OSCC underscores the need for better prognostic tools, as recently reported in the 8th edition of American Joint Committee on Cancer staging system [1].
[1] Mascitti M, et al. American Joint Committee on Cancer staging system 7th edition versus 8th edition: any improvement for patients with squamous cell carcinoma of the tongue? Oral Surg Oral Med Oral Pathol Oral Radiol. 2018 Nov;126(5):415-23.
Author Response
Reviewer 2:
In this nice work, the Authors investigated the use of metabolic tumor imaging with 18-fluorodeoxyglucose positron emission tomography (FDG-PET) to predict the bone invasion in OSCC. In particular, clinical, radiological and metabolic tumor imaging factors were assessed in 84 patients, using pathological data as gold standard. The maximum standardized uptake value (SUVmax), calculated using the formula: max. activity in region / (injected dose × body weight).
The results were interesting, showing that a value of SUVmax below 9.5 ruled out bone invasion preoperatively with a high specificity (97.6%).
The techniques used were appropriate and described with plenty details. Overall, this is a well-designed study with rigorous methods. The discussion is well-balanced, and the statements are supported by the data. The study is on a timely subject in view of increasing interest about the identification of new prognostic and predictive markers in oral oncology.
Only minor language corrections should be necessary.
Furthermore, I suggest to expand the Introduction section by adding some considerations related to the epidemiology and prognosis of OSCC. In particular, the unchanging survival in patients with OSCC underscores the need for better prognostic tools, as recently reported in the 8th edition of American Joint Committee on Cancer staging system [1].
[1] Mascitti M, et al. American Joint Committee on Cancer staging system 7th edition versus 8th edition: any improvement for patients with squamous cell carcinoma of the tongue? Oral Surg Oral Med Oral Pathol Oral Radiol. 2018 Nov;126(5):415-23.
Thank you for your positive review. Spell check was repeated and language was checked again.
The introduction was extended to add some comments of epidemiology and prognosis of OSCC, with emphasis on the unchanging survival and need for predictors.
